# TSInsight: A local-global attribution framework for interpretability in time-series data

## Abstract

With the rise in employment of deep learning methods in safety-critical scenarios, interpretability is more essential than ever before. Although many different directions regarding interpretability have been explored for visual modalities, time-series data has been neglected with only a handful of methods tested due to their poor intelligibility. We approach the problem of interpretability in a novel way by proposing TSInsight[1] where we attach an auto-encoder with a sparsity-inducing norm on its output to the classifier and fine-tune it based on the gradients from the classifier and a reconstruction penalty. The auto-encoder learns to preserve features that are important for the prediction by the classifier and suppresses the ones that are irrelevant i.e. serves as a feature attribution method to boost interpretability. In other words, we ask the network to only reconstruct parts which are useful for the classifier i.e. are correlated or causal for the prediction. In contrast to most other attribution frameworks, TSInsight is capable of generating both instance-based and model-based explanations. We evaluated TSInsight along with other commonly used attribution methods on a range of different time-series datasets to validate its efficacy. Furthermore, we analyzed the set of properties that TSInsight achieves out of the box including adversarial robustness and output space contraction. The obtained results advocate that TSInsight can be an effective tool for the interpretability of deep time-series models.

## 1 Introduction

Deep learning models have been at the forefront of technology in a range of different domains including image classification (Krizhevsky et al., 2012), object detection (Girshick, 2015), speech recognition (Dahl et al., 2010), text recognition (Breuel, 2008), image captioning (Karpathy & Fei-Fei, 2015) and pose estimation (Cao et al., 2018). These models are particularly effective in automatically discovering useful features. However, this automated feature extraction comes at the cost of lack of transparency of the system. Therefore, despite these advances, their employment in safety-critical domains like finance (Knight, 2017), self-driving cars (Kim et al., 2018) and medicine (Zintgraf et al., 2017) is limited due to the lack of interpretability of the decision made by the network.

Numerous efforts have been made for the interpretation of these black-box models. These efforts can be mainly classified into two separate directions. The first set of strategies focuses on making the network itself interpretable by trading off some performance. These strategies include Self-Explainable Neural Network (SENN) (Alvarez-Melis & Jaakkola, 2018) and Bayesian non-parametric regression models (Guo et al., 2018). The second set of strategies focuses on explaining a pretrained model i.e. they try to infer the reason for a particular prediction. These attribution techniques include saliency map (Yosinski et al., 2015) and layer-wise relevance propagation (Bach et al., 2015). However, all of these methods have been particularly developed and tested for visual modalities which are directly intelligible for humans. Transferring methodologies developed for visual modalities to time-series data is difficult due to the non-intuitive nature of time-series. Therefore, only a handful of methods have been focused on explaining time-series models in the past (Kumar et al., 2017; Siddiqui et al., 2019).

We approach the attribution problem in a novel way by attaching an auto-encoder on top of the classifier. The auto-encoder is fine-tuned based on the gradients from the classifier. Rather than

---

[1]Code along with the trained models will be made publicly available upon publication

asking the auto-encoder to reconstruct the whole input, we ask the network to only reconstruct parts which are useful for the classifier i.e. are correlated or causal for the prediction. In order to achieve this, we introduce a sparsity inducing norm onto the output of the auto-encoder. In particular, the contributions of this paper are twofold:

- A novel attribution method for time-series data which makes it much easier to interpret the decision of any deep learning model. The method also leverages dataset-level insights when explaining individual decisions in contrast to other attribution methods.

- Detailed analysis of the information captured by different attribution techniques using a simple suppression test on a range of different time-series datasets. This also includes analysis of the different out of the box properties achieved by TSInsight including generic applicability, contraction in the output space and resistance against trivial adversarial noise.

## 2  RELATED WORK

Since the resurgence of deep learning in 2012 after a deep network comprehensively outperformed its feature engineered counterparts (Krizhevsky et al., 2012) on the ImageNet visual recognition challenge comprising of 1.2 million images (Russakovsky et al., 2015), deep learning has been integrated into a range of different applications to gain unprecedented levels of improvement. Significant efforts have been made in the past regarding the interpretability of deep models, specifically for image modality. These methods are mainly categorized into two different streams where the first stream is focused on explaining the decisions of a pretrained network which is much more applicable in the real-world. The second stream is directed towards making models more interpretable by trading off accuracy.

The first stream for explainable systems which attempts to explain pretrained models using attribution techniques has been a major focus of research in the past years. The most common strategy is to visualize the filters of the deep model (Zeiler & Fergus, 2013; Simonyan et al., 2013; Yosinski et al., 2015; Palacio et al., 2018; Bach et al., 2015). This is very effective for visual modalities since images are directly intelligible for humans. Zeiler & Fergus (2013) introduced deconvnet layer to understand the intermediate representations of the network. They not only visualized the network, but were also able to improve the network based on these visualizations to achieve state-of-the-art performance on ImageNet (Russakovsky et al., 2015). Simonyan et al. (2013) proposed a method to visualize class-specific saliency maps. Yosinski et al. (2015) proposed a visualization framework for image based deep learning models. They tried to visualize the features that a particular filter was responding to by using regularized optimization. Instead of using first-order gradients, Bach et al. (2015) introduced a Layer-wise Relevance Propagation (LRP) framework which identified the relevant portions of the image by distributing the contribution to the incoming nodes. Smilkov et al. (2017) introduced the SmoothGrad method where they computed the mean gradients after adding small random noise sampled from a zero-mean Gaussian distribution to the original point. Integrated gradients method introduced by Sundararajan et al. (2017) computed the average gradient from the original point to the baseline input (zero-image in their case) at regular intervals. Guo et al. (2018) used Bayesian non-parametric regression mixture model with multiple elastic nets to extract generalizable insights from the trained model. Either these methods are not directly applicable to time-series data, or are inferior in terms of intelligibility for time-series data.

Palacio et al. (2018) introduced yet another approach to understand a deep model by leveraging auto-encoders. After training both the classifier and the auto-encoder in isolation, they attached the auto-encoder to the head of the classifier and fine-tuned only the decoder freezing the parameters of the classifier and the encoder. This transforms the decoder to focus on features which are relevant for the network. Applying this method directly to time-series yields no interesting insights (Fig. 1b) into the network's preference for input. Therefore, this method is strictly a special case of the TSInsight's formulation.

In the second stream for explainable systems, Alvarez-Melis & Jaakkola (2018) proposed Self-Explaining Neural Networks (SENN) where they learn two different networks. The first network is the concept encoder which encodes different concepts while the second network learns the weightings of these concepts. This transforms the system into a linear problem with a set of features making it easily interpretable for humans. SENN trade-offs accuracy in favor of interpretability. Kim et al. (2018) attached a second network (video-to-text) to the classifier which was responsible for the

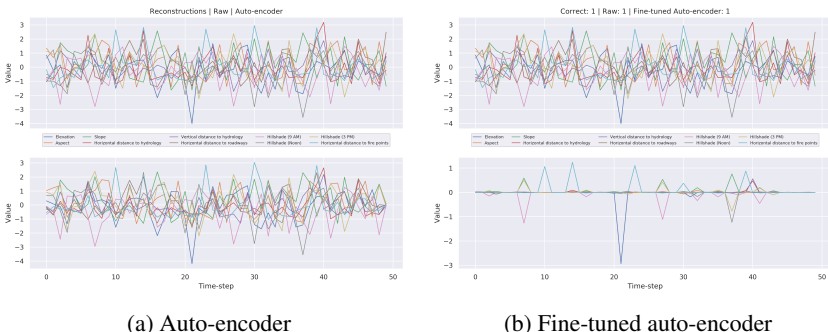

(a) Auto-encoder                   (b) Fine-tuned auto-encoder

Figure 1: Plot at the top shows the original input while plot at the bottom indicates the reconstructed input from the raw auto-encoder or the fine-tuned auto-encoder.

production of natural language based explanation of the decisions taken by the network using the saliency information from the classifier. This framework relies on LSTM for the generation of the descriptions adding yet another level of opaqueness making it hard to decipher whether the error originated from the classification network or from the explanation generator.

Kumar et al. (2017) made the first attempt to understand deep learning models for time-series analysis where they specifically focused on financial data. They computed the input saliency based on the first-order gradients of the network. Siddiqui et al. (2019) proposed an influence computation framework which enabled exploration of the network at the filter level by computing the per filter saliency map and filter importance again based on first-order gradients. However, both methods lack in providing useful insights due to the noise inherent to first-order gradients. Another major limitation of saliency based methods is the sole use of local information. Therefore, TSInsight significantly supersedes in the identification of the important regions of the input using a combination of both local information for that particular example along with generalizable insights extracted from the entire dataset in order to reach a particular description.

Due to the use of auto-encoders, TSInsight is inherently related to sparse (Ng et al., 2011) and contractive auto-encoders (Rifai et al., 2011). In sparse auto-encoders (Ng et al., 2011), the sparsity is induced on the hidden representation by minimizing the KL-divergence between the average activations and a hyperparameter which defines the fraction of non-zero units. This KL-divergence is a necessity for sigmoid-based activation functions. However, in our case, the sparsity is induced directly on the output of the auto-encoder, which introduces a contraction on the input space of the classifier, and can directly be achieved by using Manhattan norm on the activations as we obtain real-valued outputs. Albeit sparsity being introduced in both cases, the sparsity in the case of sparse auto-encoders is not useful for interpretability. In the case of contractive auto-encoders (Rifai et al., 2011), a contraction mapping is introduced by penalizing the Fobenius norm of the Jacobian of the encoder along with the reconstruction error. This makes the learned representation invariant to minor perturbations in the input. TSInsight on the other hand, induces a contraction on the input space for interpretability, thus, favoring sparsity inducing norm.

## 3 METHOD

We first train an auto-encoder as well as a classifier in isolation on the desired dataset. Once both the auto-encoder as well as the classifier are trained, we attach the auto-encoder to the head of the classifier. TSInsight is based on a novel loss formulation, which introduces a sparsity-inducing norm on the output of the auto-encoder along with a reconstruction and classification penalty for the optimization of the auto-encoder keeping the classifier fixed. Inducing sparsity on the auto-encoder's output forces the network to only reproduce relevant regions of the input to the classifier since the auto-encoder is optimized using the gradients from the classifier. As inducing sparsity on the auto-encoder's output significantly hampers the auto-encoder's ability to reconstruct the input which can in turn result in fully transformed outputs, it is important to have a reconstruction penalty in place. This effect is illustrated in Fig. 2a where the auto-encoder produced a novel sparse representation of

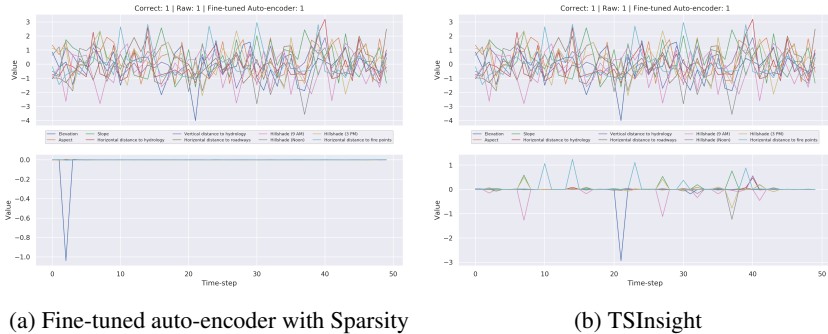

(a) Fine-tuned auto-encoder with Sparsity        (b) TSInsight

Figure 2: Plot at the top shows the original input while plot at the bottom indicates the reconstructed input either from TSInsight or the auto-encoder with just sparsity inducing norm.

the input, which albeit being an interesting one, doesn't help with the interpretability of the model. Therefore, the proposed optimization objective can be written as:

$$(\mathcal{W}'_E, \mathcal{W}'_D) = \underset{\mathcal{W}^*_E, \mathcal{W}^*_D}{\arg\min} \frac{1}{|\mathcal{X}|} \sum_{(\mathbf{x},y) \in \mathcal{X} \times \mathcal{Y}} \left[ \mathcal{L}\Big(\Phi\big(D(E(\mathbf{x};\mathcal{W}^*_E);\mathcal{W}^*_D);\mathcal{W}^*\big), y\Big) + \right.$$

$$\left. \gamma\Big(\|\mathbf{x} - D\big(E(\mathbf{x};\mathcal{W}^*_E);\mathcal{W}^*_D\big)\|_2^2\Big) + \beta\Big(\|D\big(E(\mathbf{x};\mathcal{W}^*_E);\mathcal{W}^*_D\big)\|_1\Big) \right] + \lambda\big(\|\mathcal{W}^*_E\|_2^2 + \|\mathcal{W}^*_D\|_2^2\big) \quad (1)$$

where $\mathcal{L}$ represents the classification loss function which is cross-entropy in our case, $\Phi$ denotes the classifier with pretrained weights $\mathcal{W}^*$, while $E$ and $D$ denotes the encoder and decoder respectively with corresponding pretrained weights $\mathcal{W}^*_E$ and $\mathcal{W}^*_D$. We introduce two new hyperparameters, $\gamma$ and $\beta$. $\gamma$ controls the auto-encoder's focus on reconstruction of the input. $\beta$ on the other hand, controls the sparsity enforced on the output of the auto-encoder. Pretrained weights are obtained by training the auto-encoder as well as the classifier in isolation as previously mentioned. With this new formulation, the output of the auto-encoder is both sparse as well as aligned with the input as evident from Fig. 2b.

The selection of $\beta$ can significantly impact the output of the model. Performing grid search to determine this value is not possible as large values of $\beta$ results in models which are more interpretable but inferior in terms of performance, therefore, presenting a trade-off between performance and interpretability which is difficult to quantify. A rudimentary way which we tested for automated selection of these hyperparameters ($\beta$ and $\gamma$) is via feature importance measures (Siddiqui et al., 2019; Vidovic et al., 2016). The simplest candidate for this importance measure is saliency. This can be written as:

$$I(\mathbf{x}) = \frac{\partial a^L}{\partial \mathbf{x}}$$

where $L$ denotes the number of layers in the classifier and $a^L$ denotes the activations of the last layer in the classifier. This computation is just based on the classifier i.e. we ignore the auto-encoder at this point. Once the values of the corresponding importance metric is evaluated, the values are scaled in the range of [0, 1] to serve as the corresponding reconstruction weight i.e. $\gamma$. The inverted importance values serve as the corresponding sparsity weight i.e. $\beta$.

$$I(\mathbf{x}) = \frac{I(\mathbf{x}) - \min_j I(\mathbf{x})_j}{\max_j I(\mathbf{x})_j - \min_j I(\mathbf{x})_j}$$

$$\gamma^*(\mathbf{x}) = I(\mathbf{x}) \qquad \& \qquad \beta^*(\mathbf{x}) = 1.0 - I(\mathbf{x})$$

Table 1: Dataset Details

| Dataset | Train | Validation | Test | Sequence Length | Input Channels | Output Classes |
|---|---|---|---|---|---|---|
| Synthetic Anomaly Detection | 45000 | 5000 | 10000 | 50 | 3 | 2 |
| Electric Devices | 6244 | 2682 | 7711 | 50 | 3 | 7 |
| Character Trajectories | 1383 | 606 | 869 | 206 | 3 | 20 |
| FordA | 2520 | 1081 | 1320 | 500 | 1 | 2 |
| Forest Cover | 107110 | 45906 | 65580 | 50 | 10 | 2 |
| ECG Thorax | 1244 | 556 | 1965 | 750 | 1 | 42 |
| WESAD | 5929 | 846 | 1697 | 700 | 8 | 3 |
| UWave Gesture | 624 | 272 | 3582 | 946 | 1 | 8 |

Therefore, the final term imposing sparsity on the classifier can be written as:

$$\gamma\Big(\|\mathbf{x} - D\big(E(\mathbf{x}; \mathcal{W}_E^*); \mathcal{W}_D^*\big)\|_2^2\Big) + \beta\Big(\|D\big(E(\mathbf{x}; \mathcal{W}_E^*); \mathcal{W}_D^*\big)\|_1\Big) \Rightarrow$$

$$C \times \|D\big(E(\mathbf{x}; \mathcal{W}_E^*); \mathcal{W}_D^*\big) \odot \beta^*(\mathbf{x})\|_1 + \|\Big(\mathbf{x} - D\big(E(\mathbf{x}; \mathcal{W}_E^*); \mathcal{W}_D^*\big)\Big) \odot \gamma^*(\mathbf{x})\|_2^2$$

In contrast to the instance-based value of $\beta$, we used the average saliency value in our experiments. This ensures that the activations are not sufficiently penalized so as to significantly impact the performance of the classifier. Due to the low relative magnitude of the sparsity term, we scaled it by a constant factor $C$ (we used $C = 10$ in our experiments). This approach despite being interesting, still results in inferior performance as compared to manual fine-tuning of hyperparameters. This needs further investigation for it to work in the future.

## 4 DATASETS

In order to investigate the efficacy of TSInsight, we employed several different time-series datasets in this study. The summary of the datasets is available in Table 1.

**Synthetic Anomaly Detection Dataset:** The synthetic anomaly detection dataset (Siddiqui et al., 2019) is a synthetic dataset comprising of three different channels referring to the pressure, temperature and torque values of a machine running in a production setting where the task is to detect anomalies. The dataset only contains point-anomalies. If a point-anomaly is present in a sequence, the whole sequence is marked as anomalous. Anomalies were intentionally never introduced on the pressure signal in order to identify the treatment of the network to that particular channel.

**Electric Devices Dataset:** The electric devices dataset (Hills et al., 2014) is a small subset of the data collected as part of the UK government's sponsored study, *Powering the Nation*. The aim of this study was to reduce UK's carbon footprint. The electric devices dataset is comprised of data from 251 households, sampled in two-minute intervals over a month.

**Character Trajectories Dataset:** The character trajectories dataset[2] contains hand-written characters using a Wacom tablet. Only three dimensions are kept for the final dataset which includes x, y and pen-tip force. The sampling rate was set to be 200 Hz. The data was numerically differentiated and Gaussian smoothen with $\sigma = 2$. The task is to classify the characters into 20 different classes.

**FordA Dataset:** The FordA dataset[3] was originally used for a competition organized by IEEE in the IEEE World Congress on Computational Intelligence (2008). It is a binary classification problem where the task is to identify whether a certain symptom exists in the automotive subsystem. FordA dataset was collected with minimal noise contamination in typical operating conditions.

**Forest Cover Dataset:** The forest cover dataset (Tan et al., 2011) has been adapted from the UCI repository for the classification of forest cover type from cartographic variables. The dataset has been transformed into an anomaly detection dataset by selecting only 10 quantitative attributes out of a total of 54. Instances from the second class were considered to be normal while instances from the

---

[2] https://archive.ics.uci.edu/ml/datasets/Character+Trajectories
[3] http://www.timeseriesclassification.com/description.php?Dataset=FordA

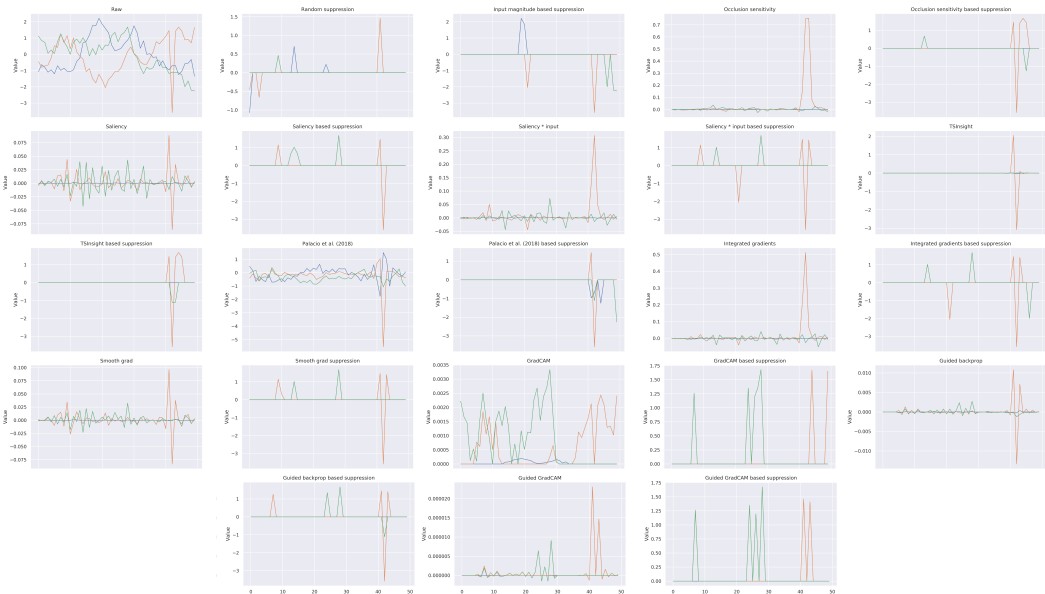

Figure 3: Output from different attribution methods as well as the input after suppressing all the points except the top 5% highlighted by the corresponding attribution method on an anomalous example from the synthetic anomaly detection dataset.

fourth class were considered to be anomalous. The ratio of the anomalies to normal data points is 0.90%. Since only two classes were considered, the rest of them were discarded.

**WESAD Dataset:** WESAD dataset (Schmidt et al., 2018) is a classification dataset introduced by Bosch for person's affective state classification with three different classes, namely, neutral, amusement and stress.

**ECG Thorax Dataset:** The non-invasive fetal ECG Thorax dataset[4] is a classification dataset comprising of 42 classes.

**UWave Gesture Dataset:** The wave gesture dataset (Liu et al., 2009) contains accelerometer data where the task is to recognize 8 different gestures.

## 5 MAIN RESULTS

The results we obtained with the proposed formulation were highly intelligible for the datasets we employed in this study. TSInsight produced a sparse representation of the input focusing only on the salient regions. With a careful tuning of the hyperparameters, TSInsight outperformed the base classifier in terms of accuracy for most of the cases. This is evident from Table 3 (Appendix G). However, it is important to note that TSInsight is not designed for the purpose of performance, but rather for interpretability. Therefore, we expect that the performance will drop in many cases depending on the amount of sparsity enforced.

In order to assess the obtained reconstructions qualitatively, we visualize an anomalous example from the synthetic anomaly detection dataset in Fig. 3 along with the attributions from all the commonly employed attribution techniques (listed below) including TSInsight. Since there were only a few relevant discriminative points in the case of forest cover and synthetic anomaly detection datasets, the auto-encoder suppressed most of the input making the decision directly interpretable.

A simple way to quantify the quality of the attribution is to just preserve parts of the input that are considered to be important by the method, and then pass the suppressed input to the classifier. If

---

[4]http://www.timeseriesclassification.com/description.php?Dataset=NonInvasiveFetalECGThorax1

the selected points are indeed causal for the prediction generated by the classifier, the prediction would stand. Otherwise, the prediction will flip. It is important to note that unless there is a high amount of sparsity present in the signal, suppressing the signal itself will result in a loss of accuracy for the classifier since there is a slight mismatch for the classifier for the inputs seen during training. We compared TSInsight with a range of different saliency methods. In all of the cases, we used the absolute magnitude of the corresponding feature attribution method to preserve the most-important input features. Two methods i.e. $\epsilon - LRP$ and DeepLift were shown to be similar to $input \odot gradient$ (Adebayo et al., 2018), therefore, we compare only against $input \odot gradient$. We don't compute class-specific saliency, but instead, compute the saliency w.r.t. all the output classes. For all the methods computing class specific activations maps e.g. GradCAM, guided GradCAM, and occlusion sensitivity, we used the class with the maximum predicted score as our target. Following is the list of the evaluated attribution techniques:

**None:** None refers to the absence of any importance measure. Therefore, in this case, the complete input is passed on to the classifier without any suppression for comparison.

**Random:** Random points from the input are suppressed in this case.

**Input Magnitude:** We treat the absolute magnitude of the input to be a proxy for the features' importance.

**Occlusion sensitivity:** We iterate over different input channels and positions and mask the corresponding input features with a filter size of 3 and compute the difference in the confidence score of the predicted class (i.e. the class with the maximum score on the original input). We treat this sensitivity score as the features' importance. This is a brute-force measure of feature importance and employed commonly in prior literature as served as a strong baseline in our experiments (Zeiler & Fergus, 2013).

**TSInsight:** We treat the absolute magnitude of the output from the auto-encoder of TSInsight as features' importance.

**Palacio et al.:** Similar to TSInsight, we use the absolute magnitude of the auto-encoder's output as the features' importance (Palacio et al., 2018).

**Gradient:** We use the absolute value of the raw gradient of the classifier w.r.t. to all of the classes as the features' importance (Siddiqui et al., 2019; Kumar et al., 2017).

**Gradient $\odot$ Input:** We compute the Hadamard (element-wise) product between the gradient and the input, and use its absolute magnitude as the features' importance (Sundararajan et al., 2017).

**Integrated Gradients:** We use absolute value of the integrated gradient with 100 discrete steps between the input and the baseline (which was zero in our case) as the features' importance (Sundararajan et al., 2017).

**SmoothGrad:** We use the absolute value of the smoothened gradient computed by using 100 different random noise vector sampled from a Gaussian distribution with zero mean, and a variance of $2/(max_j \mathbf{x_j} - min_j \mathbf{x_j})$ where $\mathbf{x}$ was the input as the features' importance measure (Smilkov et al., 2017).

**Guided Backpropagation:** We use the absolute value of the gradient provided by guided backpropagation (Springenberg et al., 2014). In this case, all the ReLU layers were replaced with guided ReLU layers which masks negative gradients, hence filtering out negative influences for a particular class to improve visualization.

**GradCAM:** We use the absolute value of Gradient-based Class Activation Map (GradCAM) (Selvaraju et al., 2016) as our feature importance measure. GradCAM computes the importance of the different filters present in the input in order to come up with a metric to score the overall output. Since GradCAM visualizes a class activation map, we used the predicted class as the target for visualization.

**Guided GradCAM:** Guided GradCAM Selvaraju et al. (2016) is a guided variant of GradCAM which performs a Hadamard product (pointwise) of the signal from guided backpropagation and GradCAM to obtain guided GradCAM. We again use the absolute value of the guided GradCAM output as importance measure.

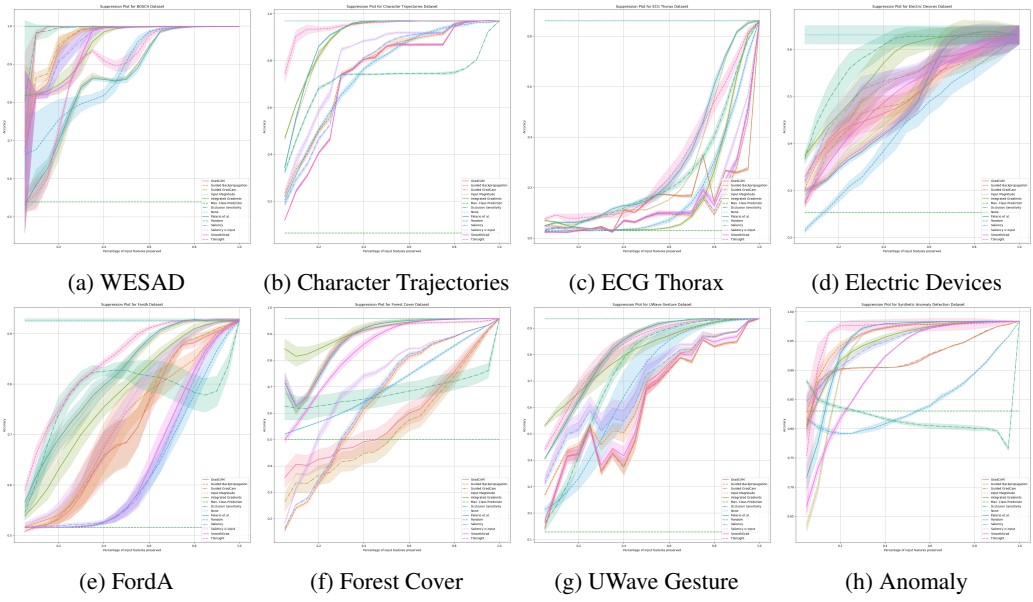

| (a) WESAD | (b) Character Trajectories | (c) ECG Thorax | (d) Electric Devices |

| (e) FordA | (f) Forest Cover | (g) UWave Gesture | (h) Anomaly |

Figure 4: Suppression results against a large number of baseline methods averaged over 5 random runs. Enlarged version of the plots is available in Appendix H.

The results with different amount of suppression are visualized in Fig. 4 which are averaged over 5 random runs. Since the datasets were picked to maximize diversity in terms of the features, there is no perfect method which can perfectly generalize to all the datasets. The different attribution techniques along with the corresponding suppressed input is visualized in Fig. 3 for the synthetic anomaly detection datasets. TSInsight produced the most plausible looking explanations along with being the most competitive saliency estimator on average in comparison to all other attribution techniques. Alongside the numbers, TSInsight was also able to produce the most plausible explanations.

## 6 PROPERTIES OF TSINSIGHT

### 6.1 GENERIC APPLICABILITY

TSInsight is compatible with any base model. We tested our method with two prominent architectural choices in time-series data i.e. CNN and LSTM. The results highlight that TSInsight was capable of extracting the salient regions of the input regardless of the underlying architecture. It was interesting to note that since LSTM uses memory cells to remember past states, the last point was found to be the most salient. For CNN on the other hand, the network had access to the complete information resulting in equal distribution of the saliency. A visual example is presented in Appendix E.

### 6.2 MODEL-BASED VS INSTANCE-BASED EXPLANATIONS

Since TSInsight poses the attribution problem itself as an optimization objective, the data based on which this optimization problem is solved defines the explanation scope. If the optimization problem is solved for the complete dataset, this tunes the auto-encoder to be a generic feature extractor, enabling extraction of model/dataset-level insights using the attribution. In contrary, if the optimization problem is solved for a particular input, the auto-encoder discovers an instance's attribution. This is contrary to most other attribution techniques which are only instance specific.

### 6.3 AUTO-ENCODER'S JACOBIAN SPECTRUM ANALYSIS

Fig. 5 visualizes the histogram of singular values of the average Jacobian on test set of the forest cover dataset. We compare the spectrum of the formulation from Palacio et al. (2018) and TSInsight. It is evident from the figure that most of the singular values for TSInsight are close to zero, indicating a

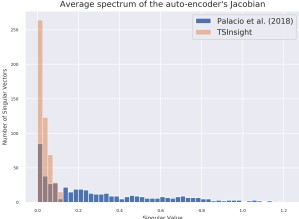

Figure 5: Spectrum analysis of the auto-encoder's average Jacobian computed over the entire test set of the forest cover dataset. Large singular values corresponds to directions which the network retained. The sharp decrease in the spectrum for TSInsight suggests that the network was successful in inducing a contraction of the input space.

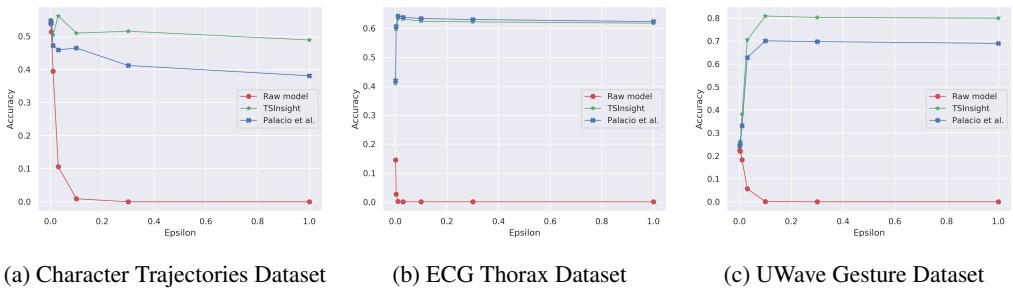

(a) Character Trajectories Dataset     (b) ECG Thorax Dataset     (c) UWave Gesture Dataset

Figure 6: Accuracy of the adversarial examples curated using FGSM attack with varying max $\epsilon$.

contraction being induced in those directions. This is similar to the contraction induced in contractive auto-encoders (Rifai et al., 2011) without explicitly regularizing the Jacobian of the encoder.

### 6.4 ADVERSARIAL ROBUSTNESS

We used the fastest and the simplest attack i.e. Fast-Gradient Sign Method (Goodfellow et al., 2014) as a dummy check. The accuracy with increasing values of $\epsilon$ are plotted to provide a hint regarding the possibility of attaining higher level of robustness since the setup is similar to high-level representation guided denoising with an emphasis on interpretability instead of robustness (Liao et al., 2018). Recent studies have indicated that the two objectives i.e. interpretability and robustness are complementary to each other (Engstrom et al., 2019). Fig. 6 indicates that TSInsight achieved a high level of immunity against adversarial noise in comparison to the base classifier which is better or on par with Palacio et al. (2018). It is important to note that this is not a proper adversarial evaluation, but rather, only a speculation which needs further investigation in the future.

## 7 CONCLUSION

We presented a novel method to discover the salient features of the input for the prediction by using the global context. With the obtained results, it is evident that the features highlighted by TSInsight are intelligible as well as reliable at the same time. In addition to interpretability, TSInsight also offers off the shelf properties which are suitable in a wide range of problems. Interpretability is essential in many domains, and we believe that our method opens up a new research direction for interpretability of deep models for time-series analysis. We would like to further investigate the automated selection of hyperparameters in the future which is primitive for the wide-scale applicability of TSInsight along with its impact on adversarial robustness.

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

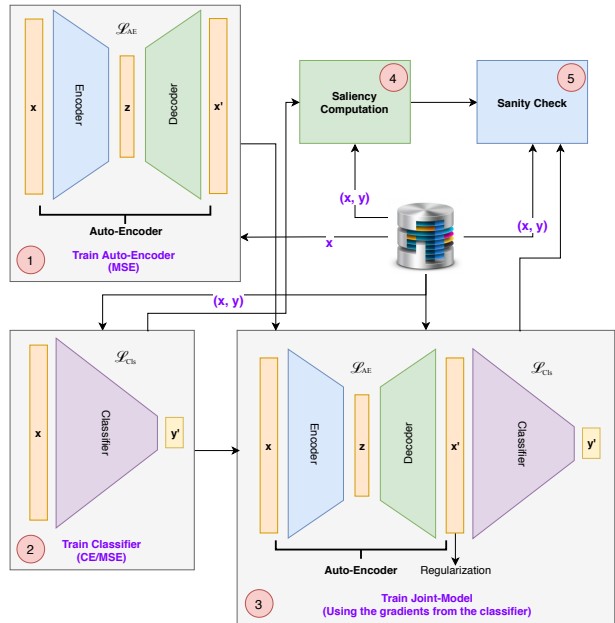

Figure 7: System Pipeline

Marina M-C Vidovic, Nico Görnitz, Klaus-Robert Müller, and Marius Kloft. Feature importance measure for non-linear learning algorithms. *arXiv preprint arXiv:1611.07567*, 2016.

Jason Yosinski, Jeff Clune, Anh Nguyen, Thomas Fuchs, and Hod Lipson. Understanding neural networks through deep visualization. In *Deep Learning Workshop, International Conference on Machine Learning (ICML)*, 2015.

Matthew D. Zeiler and Rob Fergus. Visualizing and understanding convolutional networks. *CoRR*, abs/1311.2901, 2013. URL http://arxiv.org/abs/1311.2901.

Luisa M Zintgraf, Taco S Cohen, Tameem Adel, and Max Welling. Visualizing deep neural network decisions: Prediction difference analysis. *arXiv preprint arXiv:1702.04595*, 2017.

# A APPENDIX

# B SYSTEM PIPELINE

The system overview pipeline is visualized in Fig. 7.

## C HYPERPARAMETERS

Table 2: Hyperparameters

| Hyperparameter | Value |
|---|---|
| Initial learning rate | 0.0001 |
| Learning rate reduction factor | 0.9 |
| Learning rate reduction tolerance | 4 |
| Activation regularization (L1) - $\beta$ | 0.0001 |
| Reconstruction weight - $\gamma$ | 4.0 |
| Max epochs | 50 |
| Batch size | 256 |
| Early stopping patience | 10 |

## D LOSS LANDSCAPE

We analyze the loss landscape in order to asses the impact of stacking the auto-encoder on top of the original network on the overall optimization problem. We follow the scheme suggested by Li et al. (2018) Li et al. (2017) where we first perform filter normalization using the norm of the filters. This allows the network to be scale invariant. We then sample two random directions ($\delta$ and $\eta$) and use a linear combination of these directions to identify the loss landscape. We keep the values of the classifier in the combined model intact since we treat those parameters as fixed. The function representing the manifold can be written as:

$$f(\alpha, \beta) = \mathcal{L}(\theta^* + \alpha\delta + \beta\eta) \quad \forall \alpha, \beta \in \{-1.0, -0.95, -0.90, ..., 0.90, 0.95, 1.0\} \quad (2)$$

Once the loss function is evaluated for all the values of $\alpha$ and $\beta$ (4000 different combinations), we plot the resulting function as a 3D surface. This loss landscape for the model trained on forest cover dataset is visualized in Fig. 8. The surface at the bottom (mostly in blue) signifies the loss landscape for the classifier. The landscape was nearly convex. The surface on the top is from the model coupled with the auto-encoder. It can be seen that the loss landscape has a kink at the optimal position but remains flat otherwise with a significantly higher loss value. This indicates that problem of optimizing the auto-encoder using gradients from the classifier is a significantly harder one to solve. This is consistent with our observation where the network failed to converge in many cases. Similar observations have been made by Palacio et al. Palacio et al. (2018) where they failed to fine-tune the complete auto-encoder, resorting to only fine-tuning of the decoder to make the problem tractable. The results were very similar when tested on other datasets.

## E GENERIC APPLICABILITY

An example of TSInsight trained with a CNN and an LSTM is presented in Fig. 9.

## F HARDWARE

All the models were trained on a single V-100 (16 GB) using the DGX-1 system. The system is equipped with 512 GB Ram and two Xeon E5-2698 v4 processors (2.20GHz).

## G CLASSIFICATION ACCURACY

The results from the classifier training highlighting the obtained accuracies are presented in Table 3. It is evident from the table that attaching TSInsight had no statistically significant impact on the classification performance. Although the accuracy itself went up after attaching the auto-encoder, we consider it to be a coincidence rather than a feature of TSInsight.

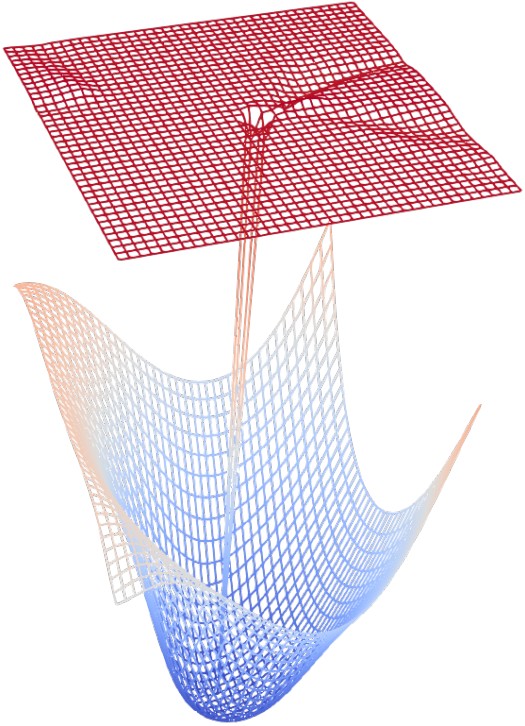

Figure 8: Loss landscape where the bottom surface indicates the manifold for the classifier while the surface on the top indicates the manifold for the auto-encoder attached to the classifier.

Table 3: Results for the different datasets in terms of accuracy for both the classifier as well as TSInsight.

| Dataset | Model | $\gamma$ | $\beta$ | Accuracy |
|---|---|---|---|---|
| Synthetic Anomaly Detection | Classifier | - | - | 98.01 % |
| | TSInsight | 1.0 | 0.001 | 98.13 % |
| WESAD | Classifier | - | - | 99.94 % |
| | TSInsight | 2.0 | 0.00001 | 99.76 % |
| Character Trajectories | Classifier | - | - | 97.01 % |
| | TSInsight | 0.25 | 0.0001 | 97.24 % |
| FordA | Classifier | - | - | 91.74 % |
| | TSInsight | 2.0 | 0.0001 | 93.26 % |
| Forest Cover | Classifier | - | - | 95.79 % |
| | TSInsight | 4.0 | 0.0001 | 96.26 % |
| Electric Devices | Classifier | - | - | 65.14 % |
| | TSInsight | 4.0 | 0.0001 | 65.74 % |
| ECG Thorax | Classifier | - | - | 86.01 % |
| | TSInsight | 0.1 | 0.0001 | 84.07 % |
| UWave Gesture | Classifier | - | - | 91.76 % |
| | TSInsight | 4.0 | 0.0005 | 92.29 % |

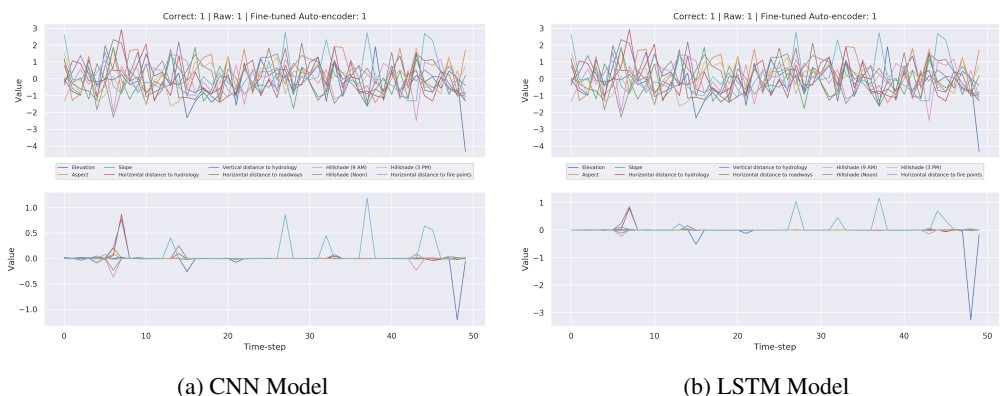

(a) CNN Model                                  (b) LSTM Model

Figure 9: Auto-encoder training with different base models (CNN and LSTM). TSInsight was able to discover salient regions of the input regardless of the employed classifier.

## H    ATTRIBUTION RESULTS

The enlarged version of the attribution plots (Fig. 4) are presented in four different figures i.e. Fig. 10, Fig. 11, Fig. 12 and Fig. 13.

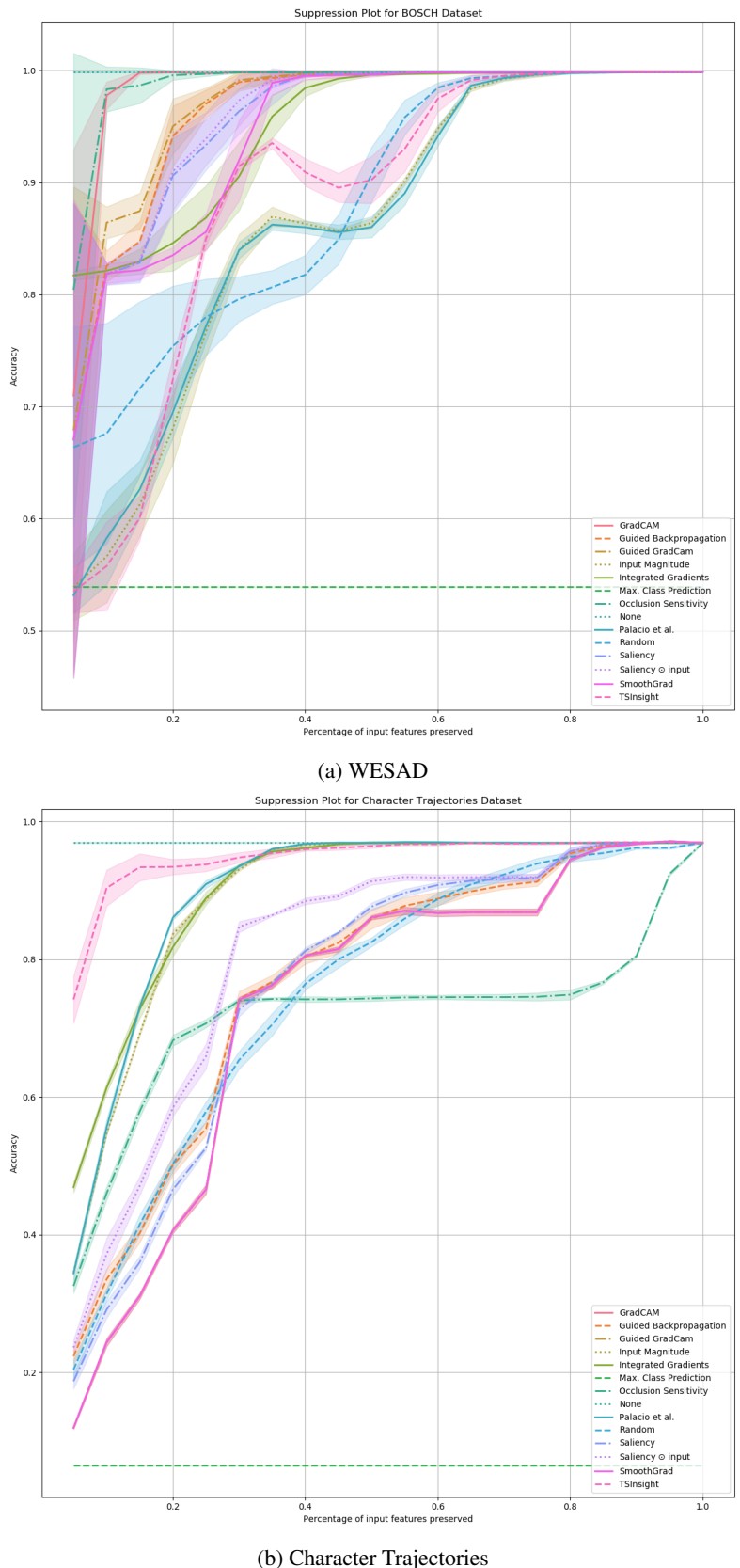

(a) WESAD

(b) Character Trajectories

Figure 10: Enlarged suppression plots (a). Copy of Fig. 4.

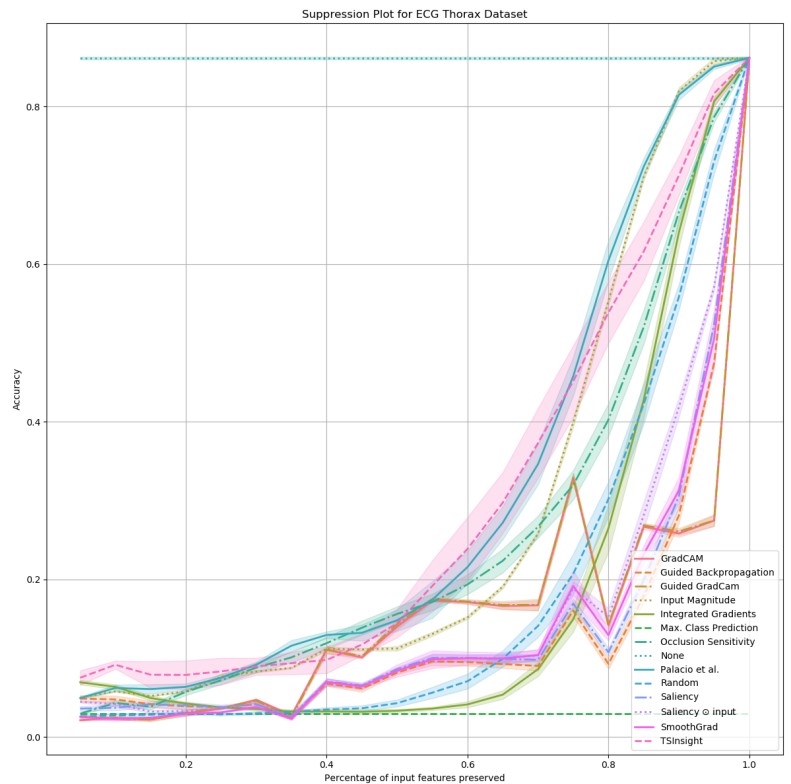

(a) ECG Thorax

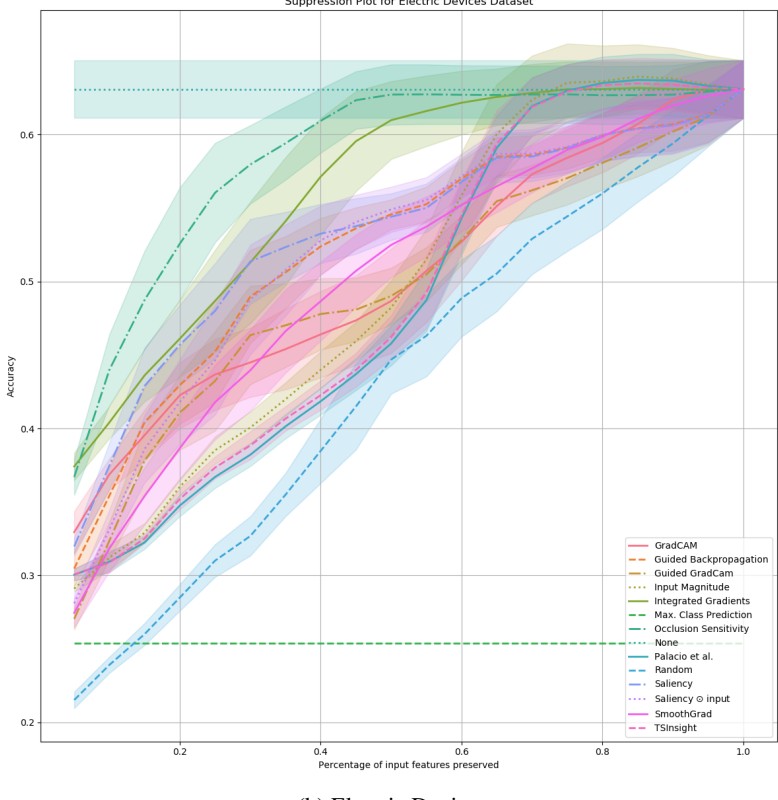

(b) Electric Devices

Figure 11: Enlarged suppression plots (b). Copy of Fig. 4.

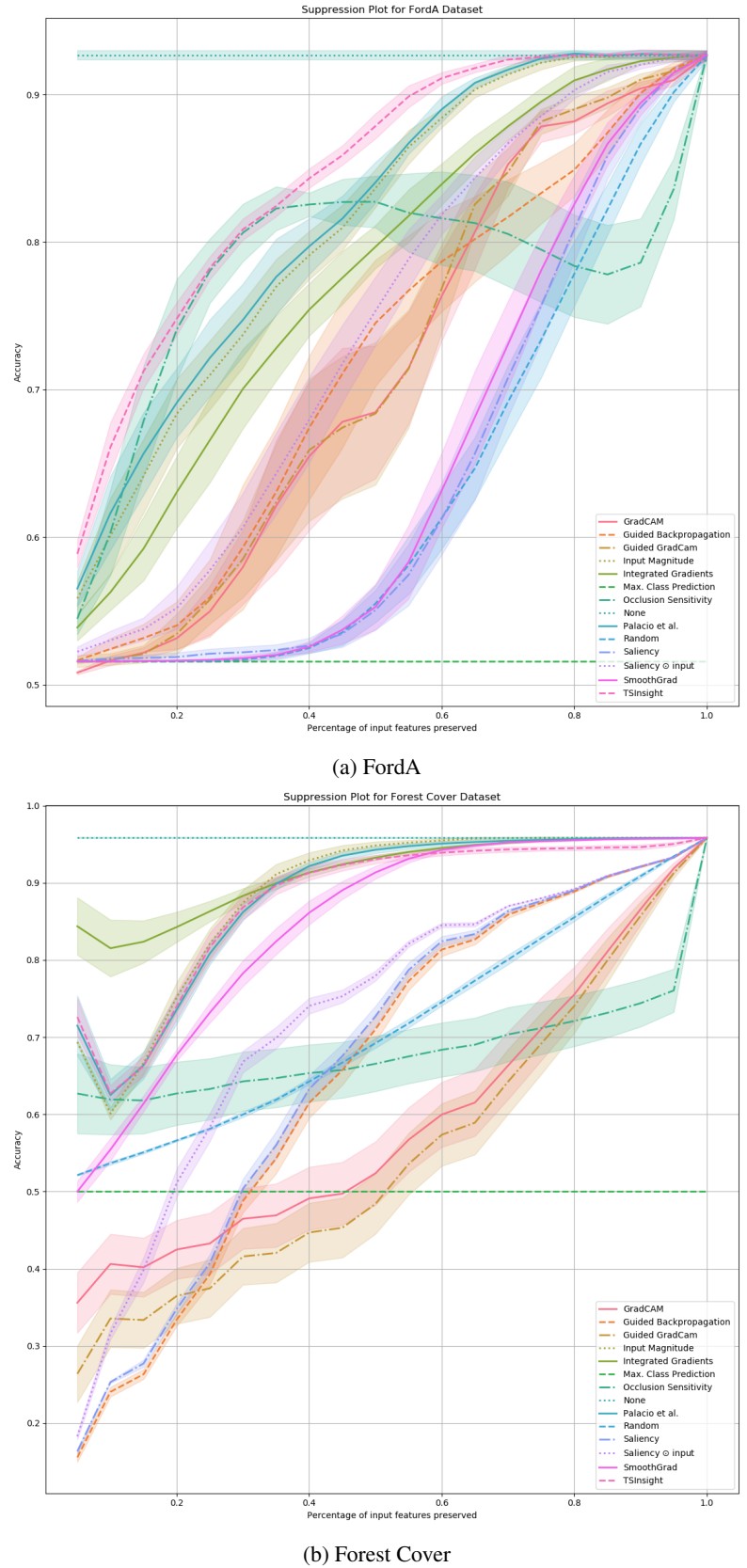

(a) FordA

(b) Forest Cover

Figure 12: Enlarged suppression plots (c). Copy of Fig. 4.

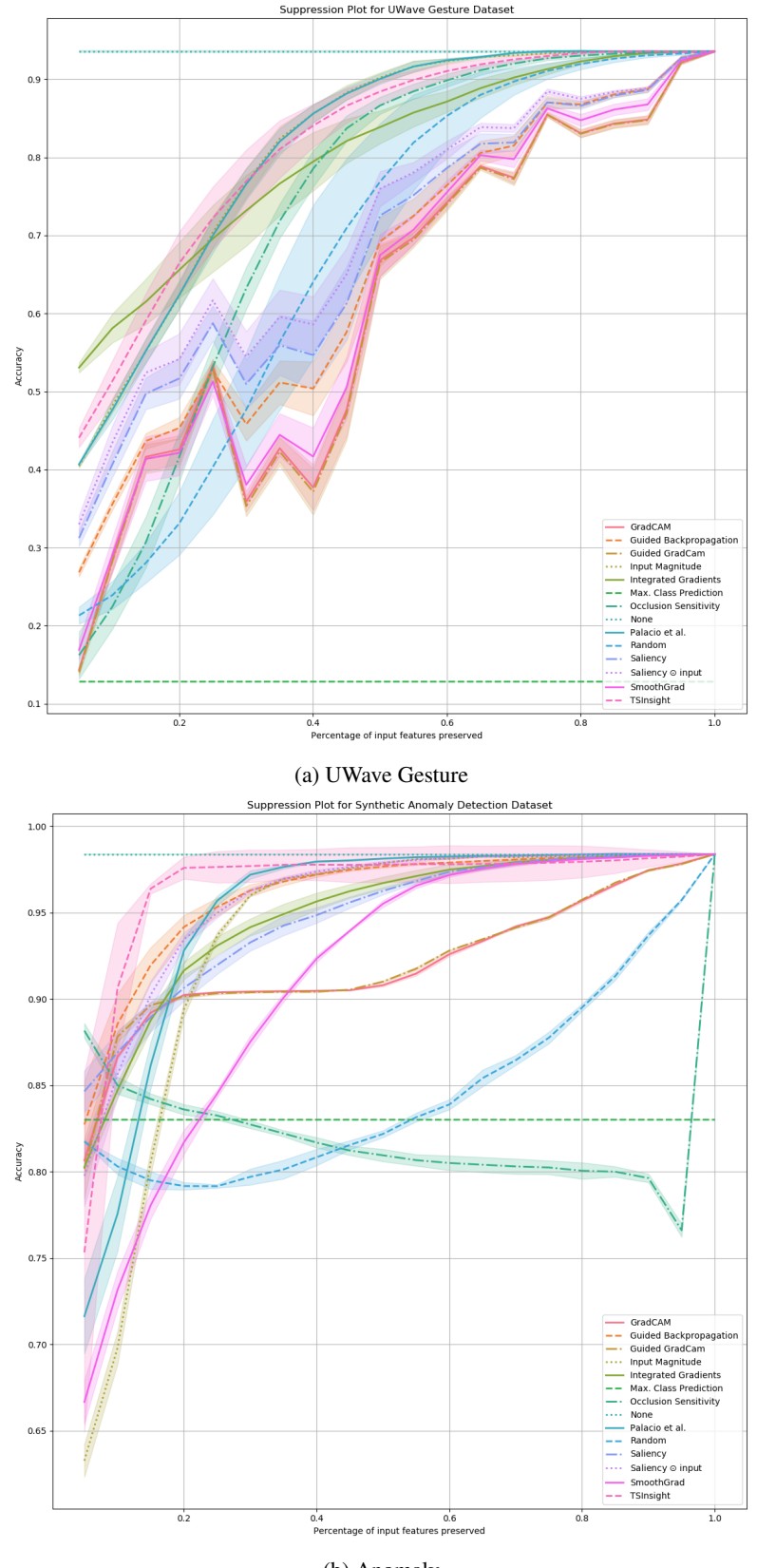

(a) UWave Gesture

(b) Anomaly

Figure 13: Enlarged suppression plots (d). Copy of Fig. 4.

