# OpenReview forum: "TSInsight: A local-global attribution framework for interpretability in time-series data"
_ICLR.cc/2020/Conference — Reject_

### Official Review · AnonReviewer1 · 2019-10-22
**Official Blind Review #1**

**Rating:** 3

**Review:**

In this paper, the authors proposed an algorithm for identifying important inputs for the time-series data as an explanation of the model's output.
Given a fixed model, the authors proposed to put an auto-encoder to the input of the model, so that the input data is first transformed through the auto-encoder, and the transformed input is then fed to the model.
In the proposed algorithm, the auto-encoder is trained so that (i) the prediction loss on the model's output to be small, (ii) the reconstruction loss of the auto-encoder to be small, and (iii) the transformed input of the auto-encoder to be sufficiently sparse (i.e. it has many zeros).

I am not very sure if the proposed algorithm can generate reasonable explanation, for the following two reasons.

First, the auto-encoder transforms the input into sparse, which can completely differ from any of the "natural" data, as shown in Fig1(b).
I am not very sure whether studying the performance of the model for such an "outlying" input is informative.

Second, it seems the authors implicitly assumed that zero input is irrelevant to the output of the model.
However, zero input can have a certain meaning to the model, and thus naively introducing the sparsity to the model input may bias the model's output.

I think the paper lacks deeper considerations on the use of sparsity.
Thus, to me, the soundness of the proposed approach is not very clear to me.


### Updated after author response ###
Through the communication with the author, I found that the author seems to be confusing the two different approaches *sparsifying the feature importance* and *sparsifying the input data*.
This paper considers the latter approach, which can be biased as I mentioned above.
The authors tried to justify the approach by raising related studies, which is however the mix of the two different approaches.
I expect the author to clarify the distinction between the two approaches, and provide strong evidences that the sparsity is not a harm (if there is any).

**Experience Assessment:**

I have read many papers in this area.

**Review Assessment: Checking Correctness Of Derivations And Theory:**

I assessed the sensibility of the derivations and theory.

**Review Assessment: Checking Correctness Of Experiments:**

I assessed the sensibility of the experiments.

**Review Assessment: Thoroughness In Paper Reading:**

I made a quick assessment of this paper.

---

> ### Author Response · Authors · 2019-11-14
> **Response to Reviewer # 01**
>
> We would like to first thank the reviewer for spending his time on the perusal of our paper.
>
> Q:
> First, the auto-encoder transforms the input into sparse, which can completely differ from any of the "natural" data, as shown in Fig1(b).
> I am not very sure whether studying the performance of the model for such an "outlying" input is informative.
>
> R:
> The attribution methods have to attribute the importance to some particular locations while discarding the rest. So it is always assumed to be sparse. The sparsity that we induce is not a random one, but is very focused from the perspective of the classifier. In order to ensure that we don’t deviate from the natural distribution, we introduced reconstruction loss into the picture. Therefore, we think that the problem is perfectly justified and entirely conforms to the prior work in this direction.
>
> Q:
> Second, it seems the authors implicitly assumed that zero input is irrelevant to the output of the model.
> However, zero input can have a certain meaning to the model, and thus naively introducing the sparsity to the model input may bias the model's output.
>
> R:
> We agree with the reviewer that the zero input can have certain meaning. Since there is no reliable test to quantify interpretability, this is very common in the literature [1] [2]. Therefore, this biasness exists in almost all of the methods that require a baseline input.
>
> [1] Fong, R., Patrick, M. and Vedaldi, A., 2019. Understanding Deep Networks via Extremal Perturbations and Smooth Masks. In Proceedings of the IEEE International Conference on Computer Vision (pp. 2950-2958).
>
> [2] Sundararajan, M., Taly, A. and Yan, Q., 2017, August. Axiomatic attribution for deep networks. In Proceedings of the 34th International Conference on Machine Learning-Volume 70 (pp. 3319-3328). JMLR. org.
>
> Q:
> I think the paper lacks deeper considerations on the use of sparsity. Thus, to me, the soundness of the proposed approach is not very clear to me.
>
> R:
> The reviewer should point out concrete instances where information is missing or incorrect so that we can clarify the confusion. “I think it lack deeper considerations” is not a well-defined argument which the authors can nullify.

---

> > ### Comment · AnonReviewer1 · 2019-11-14
> > **deeper consideration**
> >
> > I would like to thank the authors for the response.
> >
> > I raised two concerns on the use of sparsity, because I believe that the sparsity is not alway essential and can be harmful in some cases.
> >
> > [Sparsity may not be essential.]
> > For example, several attribution methods for images are actually dense (e.g. simple input gradient and its variations). A popular method SHAP also outputs dense attribution. I am not sure why and how the authors concluded that "sparsity is essential".
> >
> > [Sparsity can be harmful.]
> > As the authors mentioned in the reply, the selection of the baseline is a crucial issue in some of the attribution methods (including the current paper). I do not believe it is a reasonable justification saying "because prior studies took the same approach and so do we". If there is a potential bias, it should be clarified in the paper, and expectedly how we can avoid such a bias.
> >
> > As I raised above, sparsity may not be always essential and it can have potential drawbacks. The current paper completely misses the discussion on the justification of the use of sparsity. This is the reason why I pointed out the paper lacks "deeper consideration".
> >
> > Intepretability methods are for helping people understand models and induce the trust of the users. If people use intepretability methods without noticing potential risks (e.g. it is biased towards zero inputs), people may be mislead by the methods. Indeed, the reliability of the interpretability methods is one of the recent concerns in the field [Ref1-5]. To not mislead people and to make interpretability methods reliable, I believe the paper should clarify both pros and cons (especially what is the potential risk), rather than just saying our method is nice.
> >
> > [Ref1] A Theoretical Explanation for Perplexing Behaviors of Backpropagation-based Visualizations
> > [Ref2] Sanity Checks for Saliency Maps
> > [Ref3] Interpretation of Neural Networks is Fragile
> > [Ref4] Fairwashing: the risk of rationalization
> > [Ref5] Explanations can be manipulated and geometry is to blame

---

> > > ### Author Response · Authors · 2019-11-15
> > > **Response regarding deep consideration of sparsity**
> > >
> > > Thanks for the quick response in elaborating on your concern. We really appreciate it. Let us try to clarify.
> > >
> > > Q:
> > > [Sparsity may not be essential.]
> > > For example, several attribution methods for images are actually dense (e.g. simple input gradient and its variations). A popular method SHAP also outputs dense attribution. I am not sure why and how the authors concluded that "sparsity is essential".
> > >
> > > R:
> > > Attribution can be dense if required, however, we are not interested in the densest attribution, but rather on the most sparse attribution that still retains the prediction [1] [2] [3]. Otherwise, a trivial solution for dense attribution is just to predict the whole image to be responsible for the prediction. This is certainly correct but not useful. Therefore, we still believe that for human understanding, attributing the prediction to the smallest region possible i.e. jotting it down to the root cause is important. That is usually termed as the complexity of the explanation, and our sparsity-based framework focuses on finding the explanation with the least complexity [3].
> > >
> > > [1] Fong, R., Patrick, M. and Vedaldi, A., 2019. Understanding Deep Networks via Extremal Perturbations and Smooth Masks. In Proceedings of the IEEE International Conference on Computer Vision (pp. 2950-2958).
> > >
> > > [2] Fong, R.C. and Vedaldi, A., 2017. Interpretable explanations of black boxes by meaningful perturbation. In Proceedings of the IEEE International Conference on Computer Vision (pp. 3429-3437).
> > >
> > > [3] Ribeiro, M.T., Singh, S. and Guestrin, C., 2016, August. Why should i trust you?: Explaining the predictions of any classifier. In Proceedings of the 22nd ACM SIGKDD international conference on knowledge discovery and data mining (pp. 1135-1144). ACM.
> > >
> > > Q:
> > > [Sparsity can be harmful.]
> > > As the authors mentioned in the reply, the selection of the baseline is a crucial issue in some of the attribution methods (including the current paper). I do not believe it is a reasonable justification saying "because prior studies took the same approach and so do we". If there is a potential bias, it should be clarified in the paper, and expectedly how we can avoid such a bias.
> > >
> > > R:
> > > Having a baseline input is extremely important in interpretability and attention literature since it is important to denote the absence of a feature/input. Among all the alternates, zero input is the most plausible choice [1] [2] [3]. Sensors also use zeros to denote the absence of value. We are not sure how the reviewer thinks this can be improved. This itself can make up a seminal paper which can impact a lot of domains if a better solution exists, however, it’s hard for us to believe so. On the other hand, we think that the reviewer is overestimating the impact that the baseline input has on the generated explanation. We just want to highlight the most discriminative regions of the input discarding the rest. There are so many things wrong with the current generation of models, therefore, we believe the selection of the baseline is among the least of the concerns.
> > >
> > > As far as the references by the reviewer are concerned, we don’t have a static output problem since our model is optimized based on the classifier, so a sanity check doesn’t add any information to what is already encoded in the formulation [4]. Everything is a natural progression of the previous work in science. So we agree that the current generation of interpretability methods are not perfect, but so are the deep learning models themselves. We still have a very long way to go in that regard. We can’t have a silver bullet that can solve all the problems that are there in interpretability. We highlight this fact from the results in our paper.
> > >
> > > [1] Woo, S., Park, J., Lee, J.Y. and So Kweon, I., 2018. Cbam: Convolutional block attention module. In Proceedings of the European Conference on Computer Vision (ECCV) (pp. 3-19).
> > >
> > > [2] Sundararajan, M., Taly, A. and Yan, Q., 2017, August. Axiomatic attribution for deep networks. In Proceedings of the 34th International Conference on Machine Learning-Volume 70 (pp. 3319-3328). JMLR. org.
> > >
> > > [3] Wojna, Z., Gorban, A.N., Lee, D.S., Murphy, K., Yu, Q., Li, Y. and Ibarz, J., 2017, November. Attention-based extraction of structured information from street view imagery. In 2017 14th IAPR International Conference on Document Analysis and Recognition (ICDAR) (Vol. 1, pp. 844-850). IEEE.
> > >
> > > [4] Adebayo, J., Gilmer, J., Muelly, M., Goodfellow, I., Hardt, M. and Kim, B., 2018. Sanity checks for saliency maps. In Advances in Neural Information Processing Systems (pp. 9505-9515).

---

### Official Review · AnonReviewer3 · 2019-10-22
**Official Blind Review #3**

**Rating:** 1

**Review:**

The paper presents a new approach for improving the interpretability of deep learning methods used for time series. The is mainly concerned with classification tasks for time series. First, the classifier is learned in a usual way. Subsequently, a sparse auto-encoder is used that encodes the last layer of the classifier. For training the auto-encoder the classifier is fixed and there is a decoding loss as well as a sparsity loss. The sparse encoding of the last layer is supposed to increase the interpretability of the classification as it indicates which features are important for the classification.

In general, I think this paper needs to be considerably improved in order to justify a publication. I am not convinced about the interpretability of the sparse extracted feature vector. It is not clear to me why this should be more interpretable then other methods. The paper contains many plots where the compare to other attributes methods, but it is not clear why the presented results should be any better as other methods (for example Fig 3). The paper is missing also a lot of motivation, the results are not well explained (e.g. Fig 3) and it needs to be improved in terms of writing. Equation 1 is not motivated (which is the main part of the paper) and it is not clear how Figure 2a has been generated and why this represented should be " an interesting one, doesn’t help with the interpretability of the model". The authors have to improve the motivation part as well as the discussion of the results.

More comments below:
- The method seems to suffer from a severe parameter tuning problem, which makes it hard to use in practise.
- It is unclear to me why the discriminator is fixed during training the encoder and decoder. Shouldnt it improve performance to also adapt the discriminator to the new representation.
- Why can we not just add a layer with a sparsity constraint one layer before the "encoded" layer such that we have the same architecture  and optimize that end to end? At least comparison to such an approach would be needed to justify something more complex.
- The plots need to be better explained. While the comparisons seems to be exhaustive, it is already too many plots and it is very easy to get lost. Also, the quality of the plots need to be improved (e.g. font size)



**Experience Assessment:**

I have read many papers in this area.

**Review Assessment: Checking Correctness Of Derivations And Theory:**

I assessed the sensibility of the derivations and theory.

**Review Assessment: Checking Correctness Of Experiments:**

I assessed the sensibility of the experiments.

**Review Assessment: Thoroughness In Paper Reading:**

I read the paper at least twice and used my best judgement in assessing the paper.

---

> ### Author Response · Authors · 2019-11-14
> **Response to Reviewer # 03**
>
> We would like to first thank the reviewer for spending his time on the perusal of our paper.
>
> Q:
> The method seems to suffer from a severe parameter tuning problem, which makes it hard to use in practise.
>
> R:
> We entirely disagree with the reviewer on this since this claim is totally unbacked by the reviewer. The only reason to opt for a range of different datasets is to show that the method is generic and applicable to a wide range of different datasets. Almost all of the interpretability methods with an optimization scheme rely on hyperparameters.
>
> Q1:
> It is unclear to me why the discriminator is fixed during training the encoder and decoder. Shouldn't it improve performance to also adapt the discriminator to the new representation.
>
> Q2:
> Why can we not just add a layer with a sparsity constraint one layer before the "encoded" layer such that we have the same architecture  and optimize that end to end? At least comparison to such an approach would be needed to justify something more complex.
>
> R:
> There are two major streams of research on interpretability which we discuss in the paper. The first stream focuses on explaining the decisions of pre-trained networks while the second one focuses on making the network itself interpretable. TSInsight is particularly focused on the first steam. Therefore, TSInsight takes a pre-trained model and tries to explain the decisions made by the network using an auto-encoder with sparsity inducing norm on top of it. The classifier remains intact since we just want to explain the decisions made by the classifier rather than coming up with an architecture that is itself explainable. However, it can be easily extended for the other case as the reviewer mentioned. But it wasn’t a focus of the current work and can be explored in detail in the future.
>
> Q:
> The plots need to be better explained. While the comparisons seems to be exhaustive, it is already too many plots and it is very easy to get lost. Also, the quality of the plots need to be improved (e.g. font size)
>
> R:
> We agree that the quality of the plots as well as the accompanying text can be improved. We will work on it to make everything clear. We also included high-resolution versions of these plots in the supplementary material. The reviewer can refer to them for now in case required.

---

### Official Review · AnonReviewer2 · 2019-10-23
**Official Blind Review #2**

**Rating:** 1

**Review:**

The aim of this work is to improve interpretability in time series prediction. To do so, they propose to use a relatively post-hoc procedure which learns a sparse representation informed by gradients of the prediction objective under a trained model. In particular, given a trained next-step classifier, they propose to train a sparse autoencoder with a combined objective of reconstruction and classification performance (while keeping the classifier fixed), so as to expose which features are useful for time series prediction.  Sparsity, and sparse auto-encoders, have been widely used for the end of interpretability. In this sense, the crux of the approach is very well motivated by the literature.

* Pros
	* The work provides extensive comparison to a battery of other methods for model prediction interpretation.
	* The method is conceptually simple and is easy to implement. It is also general and can be applied to any prediction model (though this is more property of the sparse auto-encoder).
	* Despite its simplicity and generality, the method is shown to perform well on average, though it sometimes performs significantly worse than simple baselines.

* Cons
	* The method itself is not explained very well. The authors use language such as “attach the auto encoder to the classifier”, which is a bit vague and could mean a number of things. It would be helpful if they provided either a formal definition of the model or a architectural diagram.
	* Though the quantitative evaluation is not entirely flattering, the authors should not be punished for providing experiments on many datasets. That said, if their contribution is then rather one of technical novelty, i.e. a sparse-autoencoder-based framework for time series interpretability, it would be helpful for them to
		* More formally define their framework / class of solutions
		* Provide a more in depth study of possible variants of the method (this is elaborated on in the “Questions” section)
		* More strongly argue the novelty of their method
	* The authors provide a discussion on automatic hyper-parameter tuning that seems a bit out of place in the main method section, since it is not mentioned much thereafter and is claimed to not bring benefits.
	* The qualitative evaluation made by authors is rather vague:
		* "Alongside the numbers, TSInsight was also able to produce the most plausible explanations”

* Additional Remarks
	* Why not train things jointly? Does this have to be done post-hoc? The authors state that they “should expect a drop in performance since the input distribution changes” -> so why not at least try fine-tune and study the effect of training the classifier with sparse representations end-to-end? Exploring whether things can be trained jointly, or in other configurations, might allow the authors to frame their work as more of a general technical contribution.
	* It would be nice to have the simple baseline of a classifier with a sparsity constraint, i.e.
		* I.e. ablate the reconstruction loss

I’ve given a reject because 1) the explanation of the method is not very precise and could be greatly improved, 2) the quantitative evaluation is not sufficiently convincing, given the lack of technical novelty), and 3) the qualitative evaluation is hand-wavy.

**Experience Assessment:**

I have read many papers in this area.

**Review Assessment: Checking Correctness Of Derivations And Theory:**

I carefully checked the derivations and theory.

**Review Assessment: Checking Correctness Of Experiments:**

I assessed the sensibility of the experiments.

**Review Assessment: Thoroughness In Paper Reading:**

I read the paper at least twice and used my best judgement in assessing the paper.

---

> ### Author Response · Authors · 2019-11-14
> **Response to Reviewer # 02**
>
> We would like to first thank the reviewer for spending his time on the perusal of our paper.
>
> Q:
> The method itself is not explained very well. The authors use language such as “attach the auto encoder to the classifier”, which is a bit vague and could mean a number of things. It would be helpful if they provided either a formal definition of the model or an architectural diagram.
>
> R:
> We tried to elaborate on what we meant by this throughout the paper. However, as the reviewer highlighted, a diagram is quite useful to avoid confusion. We have such a diagram for TSInsight which we moved to the supplementary material due to space constraints. We can adjust it back to the main text if the reviewers find it useful for the overall idea.
>
> Q:
> Though the quantitative evaluation is not entirely flattering, the authors should not be punished for providing experiments on many datasets. That said, if their contribution is then rather one of technical novelty, i.e. a sparse-autoencoder-based framework for time series interpretability, it would be helpful for them to …
>
> R:
> We spent more space than any other paper to clearly outline the previous work in this direction and how TSInsight is technically a novel solution to this problem. It is very easy to misguide the reviewer by just adding flattering cases. It is quite common in interpretability literature to selectively pick datasets where the method shines and avoid comparison against strong baselines. However, the reason for providing a detailed comparative study on a range of different datasets and almost all the commonly employed interpretability techniques is to show that although TSInsight provides the most plausible explanations on average, there is still a very big room for improvement.
>
> Q:
> The authors provide a discussion on automatic hyper-parameter tuning that seems a bit out of place in the main method section, since it is not mentioned much thereafter and is claimed to not bring benefits.
>
> R:
> Automated hyperparameter tuning is an important avenue for this work. However, we weren’t able to obtain any interesting results through it. It is mainly intended to provide a future direction for the work.
>
> Q:
> The qualitative evaluation made by authors is rather vague, "Alongside the numbers, TSInsight was also able to produce the most plausible explanations”.
>
> R:
> We didn’t intend to provide any accompanying text for that. The qualitative evaluation was based on the plots included for the user’s perusal as common in interpretability literature.
>
> Q:
> Why not train things jointly? Does this have to be done post-hoc? The authors state that they “should expect a drop in performance since the input distribution changes” -> so why not at least try fine-tune and study the effect of training the classifier with sparse representations end-to-end? Exploring whether things can be trained jointly, or in other configurations, might allow the authors to frame their work as more of a general technical contribution.
>
> R:
> There are two major streams of research on interpretability which we discuss in the paper. The first stream focuses on explaining the decisions of pre-trained networks while the second one focuses on making the network itself interpretable. TSInsight is particularly focused on the first steam. Therefore, TSInsight takes a pre-trained model and tries to explain the decisions made by the network using an auto-encoder with sparsity inducing norm on top of it. The classifier remains intact since we just want to explain the decisions made by the classifier rather than coming up with an architecture that is itself explainable. However, it can be easily extended for the other case as the reviewer mentioned. But it wasn’t a focus of the current work and can be explored in detail in the future.
>
> Q:
> It would be nice to have the simple baseline of a classifier with a sparsity constraint, i.e. ablate the reconstruction loss
>
> R:
> We already show in the paper that removing the reconstruction loss destroys the method’s utility as an interpretability scheme. Since the aim of this work is to achieve interpretability, it doesn’t make sense to consider that as a baseline since the model won’t be providing attribution information.
>
> Overall:
> We agree that there is great room for improvement in terms of portraying the idea. However, we are quite disappointed with the reviewer’s comment on the lack of novelty and unconvincing quantitative evaluation. We compare against all the recent works in this direction and show that our method is the best on average. However, we would like to make it clear that there is no one method in the literature until this point that can provide reasonable explanations on any provided dataset. Since interpretability is itself a very tough domain to verify, we believe that our work provides one of the best coverage regarding the different attribution methods and their performance on a wide range of different datasets.

---

> > ### Comment · AnonReviewer2 · 2019-11-15
> > **response from reviewer**
> >
> > 1. ** "we are quite disappointed with the reviewer’s comment on the lack of novelty and unconvincing quantitative evaluation" **
> > Just to be clear, in my original review, I acknowledged the positive aspect of the thoroughness of your quantitative experiments. I stand by my comment that the proposed method lacks novelty.
> >
> > The sparse autoencoder is not a new model. This work applies it to the domain of interpretability. Therefore, there is limited technical novelty. That said, if your work actually proposed a general framework for using sparse autoencoders for interpretability, there could potentially be novelty in terms of formulation; however, your work completely lacks any sort of formal presentation of a formulation (not to mention the model itself). Therefore, any novelty in the formulation proposed is *not properly communicated*.
> >
> >
> > 2. ** "Automated hyperparameter tuning is an important avenue for this work. However, we weren’t able to obtain any interesting results through it. It is mainly intended to provide a future direction for the work." **
> >
> > I would argue that it then should not appear in the main method section. Again, this may be a problem of  writing and communication, as it muddles the presentation. Spend the extra space you have in the method section to actually define the formulation and method, in a way that actually conveys the novelty of the formulation you believe has merits.
> >
> > 3. ** "We didn’t intend to provide any accompanying text for that. The qualitative evaluation was based on the plots included for the user’s perusal as common in interpretability literature.”.
> >
> > Plausibility of an explanation is not something that can be derived from perusing plots. Either remove the comment about qualitative results, or justify them more rigorously.
> >
> > 4. ** "We already show in the paper that removing the reconstruction loss destroys the method’s utility as an interpretability scheme. "
> >
> > Sparsity has long been used as a way to enhance interpretability of regression models [1]. I was asking for the simple baseline, not necessarily your model without the reconstruction loss.
> >
> > [1] https://en.wikipedia.org/wiki/Lasso_(statistics)
> >
> >
> > ----
> >
> > Lack of novelty is not, by itself, sufficient ground for rejection. However, I feel that the way the ideas of this paper are currently presented are suboptimal; the formulation is not explicitly presented, the model is not explicitly defined. For me, the weakness of this submission is not the decent empirical performance of the method; rather, it is that there does not seem to be much else.

---

### Author Response · Authors · 2019-11-14
**Response to area chair**

We are quite disappointed with the allocation of reviewers. None of the reviewers that were assigned to us is an active researcher in the area of interpretability, so we find it very unfair from the ICLR team to ask them for review with their knowledge stemming from reading a few papers in this direction. We believe ICLR should improve its review process in the future to make sure only active researchers in a particular area are asked to assess the quality of the submitted work. We understand that the number of reviewers is limited. So rather than asking anyone to review, the conference should rather impose a cap on the number of submissions in order to ensure that each submission is given a proper assessment.

All of the reviewers insisted on optimizing the model end-to-end. However, we again emphasize that TSInsight is targeted towards explaining pre-existing models that already excel at the task they are trying to perform. We just intend to explain the decisions made by these networks, rather than training a network from scratch which is itself explainable.

As far as the results are concerned, there is no silver bullet in research i.e. “no free lunch theorem”. Therefore, we specifically highlighted this fact from our results by including a very diverse range of datasets regardless of their performance on any particular method. This indicated that although our method worked the best on average, the problem of interpretability still has a very long way to go. This complete picture is intentionally dropped from most of the papers in order to minimize the risk of the reviewer’s objection.

---

### Decision · Program_Chairs · 2019-12-19

**Decision:**

Reject

**Comment:**

Main content:

Blind review #2 summarizes it well:

The aim of this work is to improve interpretability in time series prediction. To do so, they propose to use a relatively post-hoc procedure which learns a sparse representation informed by gradients of the prediction objective under a trained model. In particular, given a trained next-step classifier, they propose to train a sparse autoencoder with a combined objective of reconstruction and classification performance (while keeping the classifier fixed), so as to expose which features are useful for time series prediction.  Sparsity, and sparse auto-encoders, have been widely used for the end of interpretability. In this sense, the crux of the approach is very well motivated by the literature.

--

Discussion:

All reviews had difficulties understanding the significance and novelty, which appears to have in large part arisen from the original submission not having sufficiently contextualized the motivation and strengths of the approach (especially for readers not already specialized in this exact subarea).

--

Recommendation and justification:

The reviews are uniformly low, probably due to the above factors, and while the authors' revisions during the rebuttal period have improved the objections, there are so many strong submissions that it would be difficult to justify override the very low reviewer scores.